# Incorporating temporal distribution of population-level viral load enables real-time estimation of COVID-19 transmission

Yun Lin[1,4], Bingyi Yang [1,4], Sarah Cobey [2], Eric H. Y. Lau [1,3], Dillon C. Adam[1], Jessica Y. Wong[1], Helen S. Bond[1], Justin K. Cheung[1], Faith Ho [1], Huizhi Gao[1], Sheikh Taslim Ali [1,3], Nancy H. L. Leung [1], Tim K. Tsang[1,3], Peng Wu [1,3], Gabriel M. Leung [1,3] & Benjamin J. Cowling [1,3✉]

Many locations around the world have used real-time estimates of the time-varying effective reproductive number ($R_t$) of COVID-19 to provide evidence of transmission intensity to inform control strategies. Estimates of $R_t$ are typically based on statistical models applied to case counts and typically suffer lags of more than a week because of the latent period and reporting delays. Noting that viral loads tend to decline over time since illness onset, analysis of the distribution of viral loads among confirmed cases can provide insights into epidemic trajectory. Here, we analyzed viral load data on confirmed cases during two local epidemics in Hong Kong, identifying a strong correlation between temporal changes in the distribution of viral loads (measured by RT-qPCR cycle threshold values) and estimates of $R_t$ based on case counts. We demonstrate that cycle threshold values could be used to improve real-time $R_t$ estimation, enabling more timely tracking of epidemic dynamics.

[1] WHO Collaborating Centre for Infectious Disease Epidemiology and Control, School of Public Health, Li Ka Shing Faculty of Medicine, The University of Hong Kong, Hong Kong, China. [2] Department of Ecology and Evolution, University of Chicago, Chicago, IL, USA. [3] Laboratory of Data Discovery for Health Limited, Hong Kong Science and Technology Park, New Territories, Hong Kong, China. [4] These authors contributed equally: Yun Lin, Bingyi Yang. ✉email: bcowling@hku.hk

Monitoring the transmission of an emerging infectious disease in a timely manner is crucial to evaluate the effectiveness of public health and social measures and to inform better control policies. During the coronavirus diseases 2019 (COVID-19) pandemic, real-time assessment of transmission has generally been achieved through monitoring the time-varying effective reproductive number, $R_t$. A number of statistical approaches have been developed to allow estimation of $R_t$ from time series of daily case counts either recorded by date of illness onset or by date of laboratory confirmation, or from time series of observed deaths[1,2]. Though efforts have been made to reduce the impact of lag in $R_t$ estimates[3,4], the majority of these approaches tend only to be able to estimate $R_t$ with a lag of one week or more, because COVID-19 transmission can occur prior to illness onset[5], because of delays between individuals being infected and polymerase chain reaction (PCR) detectable and/or showing symptoms (which are typically around 3–5 days for COVID-19)[6], and because of delays between illness onset and diagnosis. In Hong Kong, we estimated $R_t$ with a 7-day lag by accounting for pre-symptomatic transmission and reporting delays[7,8].

An individual infected by SARS-CoV-2 will typically experience viral load peaking around illness onset and monotonically decreasing during the following two weeks[9]. While viral loads can vary across individuals, with some shedding more than others[10,11], the mean distribution of viral loads from a group of patients measured around the time of illness onset will tend to have higher values than that of viral loads from a group of patients measured at a later time after infection[12]. Collectively, higher population-level viral loads would correlate with more infected persons being earlier in the course of infection and vice versa[13]. Viral loads can be proxied by cycle threshold (Ct) values in the real-time quantitative reverse-transcription polymerase chain reaction (RT-qPCR) assay, with lower Ct values indicative of higher viral loads.

A recent study showed that the distribution of viral loads among confirmed cases can provide inferences on transmission dynamics within populations, where population-level Ct values skewing towards lower values indicate more individuals have been recently infected, corresponding to an increasing rate of epidemic growth in the community, especially where single strain dominates[13]. The method was demonstrated in a modeling study using cross-sectional samples in a small-scale outbreak in Boston[13] while the correlation was also observed over an epidemic wave elsewhere[14]. Here, we incorporated Ct values from COVID-19 cases in Hong Kong, a location with intense surveillance and case-finding efforts, to demonstrate that including data on population viral load distributions from symptom-based surveillance could support real-time tracking of transmission.

## Results

In Hong Kong, COVID-19 cases are detected through clinical diagnosis for individuals with acute respiratory symptoms and public health surveillance for the community with a priority to people with pre-defined high exposure risks[15] (see Methods). Close contacts of confirmed cases are traced and placed into quarantine outside the home, and repeatedly tested. All laboratory-confirmed COVID-19 cases, including asymptomatic cases, are isolated within the hospital and receive multiple RT-qPCR tests during their stay. After excluding imported cases, we analyzed the first available record of Ct value (derived from RT-qPCR tests targeting E gene[16]) for each confirmed case and characterized the daily distribution of Ct values (measured by mean and skewness) that were sorted by sampling days. We included two consecutive epidemics in July–August 2020 (i.e., the third wave) and November 2020 through March 2021 (i.e., the fourth wave), which were dominated by local transmissions instead of imported cases[8,17].

A total of 8646 local COVID-19 cases were detected during periods studied, among which 77% ($n = 6700$) were symptomatic. Cases who were asymptomatic at the time of testing were more likely to be epidemiologically linked with other known cases compared to symptomatic cases (81% vs. 61%, chi-squared test $P<0.001$), suggesting they were more likely to be detected from contact tracing or from compulsory testing for populations with predefined high risks of exposures (see Methods) and could be detected earlier than symptomatic cases. Ct values were available for 96% ($n = 8268$) of local cases and included in further analyses. All included cases had not been vaccinated, as the local COVID-19 vaccination program began in late February 2021 towards the end of the period studied. Variants of concerns (VoCs) were not reported among local cases during the study period, while two separate wild-type lineages dominated the two studied waves[18].

We first examined the correlation between the distribution of daily Ct values and the local transmission dynamics[8] (measured by the incidence-based $R_t$; see Methods). The temporal Ct distribution tracked very closely the incidence-based $R_t$ over the two epidemic waves (Fig. 1; Supplementary Fig. 1a). Higher values of incidence-based $R_t$ were found when the average Ct values decreased (Spearman's correlation coefficient, $\rho = -0.79$, $P<0.001$ for the third wave and $\rho = -0.52$, $P<0.001$ for the fourth wave) and when the Ct skewed towards lower values (i.e., greater values of skewness estimates; $\rho = 0.80$, $P<0.001$ for the third wave and $\rho = 0.27$, $P<0.001$ for the fourth wave) (Fig. 1; Supplementary Table 1).

To confirm that the changes in the observed daily Ct distribution were mostly driven by the epidemic dynamics despite individual variations in viral shedding, we extrapolated the Ct value back to illness onset for symptomatic cases using the fitted association that Ct values increase 1.057 (95% confidence interval (CI): 1.050–1.063) per day after illness onset (Supplementary Fig. 2; see Methods). We found that distributions of Ct values at onset were less variable than those at sampling during the studied period (coefficient of variation for skewness: 0.37 vs. 0.80) (Supplementary Figs. 3 and 4), suggesting a relatively stable peaking level of viral loads across individuals over the course of the epidemic.

To use Ct values for real-time assessing COVID-19 transmission in the community, we fitted a log-linear regression to daily incidence-based $R_t$ on daily mean and skewness of Ct values at sampling during the third wave (i.e., training period; see Methods). We found that the distribution of Ct values explained 72% of the observed variations in incidence-based $R_t$ during the training period (Supplementary Table 2). We then applied the trained model to the daily Ct distributions in the fourth wave (i.e., testing period) to estimate $R_t$ in real time (i.e., Ct-based $R_t$). We found that the Ct-based method provided accurate real-time estimations of $R_t$ during the 7-day lagged window suffered by the conventional incidence-based $R_t$ estimation method (Fig. 2a). We found high correlations between Ct- and incidence-based $R_t$ for both training (Spearman's correlation coefficient, $\rho = 0.81$, $P<0.001$) and testing periods ($\rho = 0.48$, $P<0.001$) (Fig. 2b–d). We conducted sensitivity analyses to account for the potential impact of age on Ct distributions (Supplementary Fig. 5) and for changes in proportions of symptomatic cases (Supplementary Fig. 6) and the resulting Ct-based $R_t$, and found that the high correlation between Ct- and incidence-based $R_t$ remained.

We performed a further validation of our results by training the model using data from November to December 2020 (i.e., early stage of the fourth wave) and predicting the later stage of the fourth wave and the third wave, and found the high accuracy of

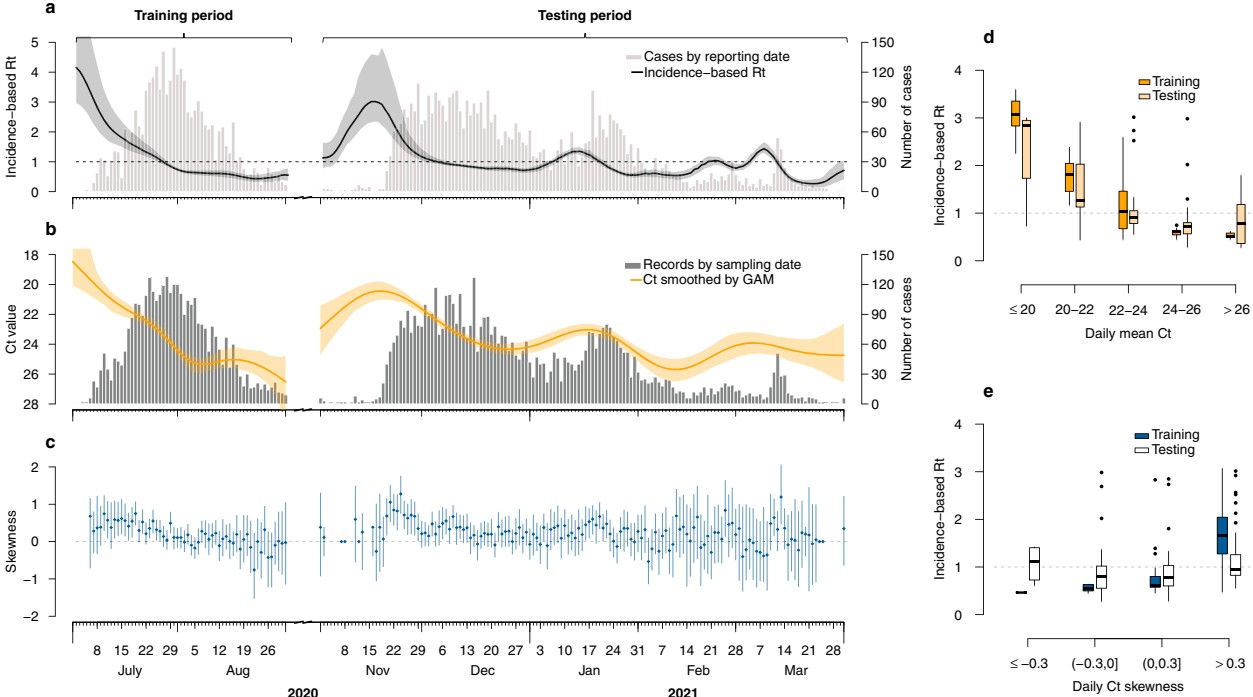

**Fig. 1 Correlations between temporal distribution of Ct values and transmission dynamics of COVID-19 in Hong Kong. a** Local COVID-19 cases and the estimated incidence-based $R_t$. Gray bars indicate the number of laboratory-confirmed local cases by date of reporting. Black lines and shaded areas indicate the mean and 95% credible intervals (CrIs) for incidence-based $R_t$. **b** Ct distributions smoothed from a generalized additive model (GAM). Dark gray bars indicate the number of sample collections. Orange lines and shaded areas indicate the daily average and 95% confidence intervals (CIs) of Ct values that were estimated from a GAM (Eq. (2)) over the study period in Hong Kong. **c** Daily skewness of Ct values over the study period. Blue dots represent the mean of daily Ct skewness and vertical lines represent 95% CIs of daily Ct skewness that were calculated from 500 bootstraps. **d, e** Correlations between the daily incidence-based $R_t$ and the daily mean Ct (**d**) or skewness (**e**). Boxes represent the interquartile range (IQR; defined as differences between 25th and 75th percentiles, same for Fig. 2) and median of the incidence-based $R_t$, lower whiskers represent the minimum and upper whiskers represent either the maximum or the largest values that are within the distance of 1.5 times the IQR of all incidence-based $R_t$ under various Ct distribution intervals, dots represent values beyond the lower and upper whiskers ($n = 59$ and 146 daily Ct mean for wave 3 and 4 in panel (**d**), and $n = 57$ and 138 daily Ct skewness for wave 3 and 4 in panel (**e**), respectively).

predictions still held (Supplementary Fig. 7). We also performed a 10-fold cross-validation, in which we randomly assigned data between 6 July 2020 to 31 March 2021 into 10 validation sets. We found that on average 81% (ranging from 75% to 85%) of the Ct- and incidence-based $R_t$ estimates were directionally consistent across validation sets. These results suggested that relationships between Ct distributions and $R_t$ estimates were not affected by temporal autocorrelation of incidence-based $R_t$. In addition, we found that our model predictions were insensitive to the selection of the training period as long as the training period had sufficient samples (e.g., >30 samples per day as suggested in Supplementary Table 4) and could reflect changes in both epidemic growth and population Ct distributions. Such training period often covered the transition point when $R_t$ shift around 1 and would span an epidemic peak in places with clear waves (Supplementary Fig. 8). Longer training periods did not necessarily lead to better performance, possibly due to the variability in the longer tail with low numbers of samples (Supplementary Figs. 1 and 8).

We used synthetic data to examine the potential impact of case detection on our methods. We simulated two consecutive epidemic waves (Supplementary Fig. 9a) using a compartment transmission model and investigated various case detection scenarios (Supplementary Fig. 9b). We also simulated individual viral load trajectories, incubation periods and sample-since-onset intervals to determine Ct values at sampling for detected symptomatic cases in the simulations (see Methods). We found that Ct-based $R_t$ can recover the simulation truth when there is limited and changing case detection (Supplementary Fig. 10).

Specifically, Ct-based $R_t$ is correlated with the simulation truth under scenarios with varying case detection (i.e., scenario 3; Spearman $\rho = 0.77$, 95% CI: 0.73–0.81) and with certain degree of under detection (i.e., scenario 4; Spearman $\rho = 0.65$, 95% CI: 0.53–0.75) (Supplementary Fig. 10c, d; Supplementary Table 7).

## Discussion

In this study, we applied a simplified Ct-based method to provide precise estimates of daily $R_t$ and demonstrated that such a method could be used for real-time $R_t$ estimation. Conventionally, the main challenge in estimating $R_t$ in real-time was largely caused by the delays between an individual being infected and being PCR detectable or illness onset[6,19]. Linking the incidence-based $R_t$ and the population-level Ct distribution among samples collected on a given day was able to mitigate the right-censoring issue (i.e., missing cases that were infected but not-yet-observed due to the latent period[6]) encountered by incidence-based methods for assessing transmission[1,6]. Although studies[3,4] demonstrated nowcasting and projection of incidence-based $R_t$ during the right-censoring time window, these estimates were indicative values rather than genuine estimates informed with real-time empirical data.

The few studies that have used population viral loads to infer COVID-19 epidemics only provided probability distributions of the estimated position of a community within an epidemic curve[13,20], while our study provides precise longitudinal $R_t$ estimates using a method that required less complicated computation efforts, which further demonstrates the potential to improve

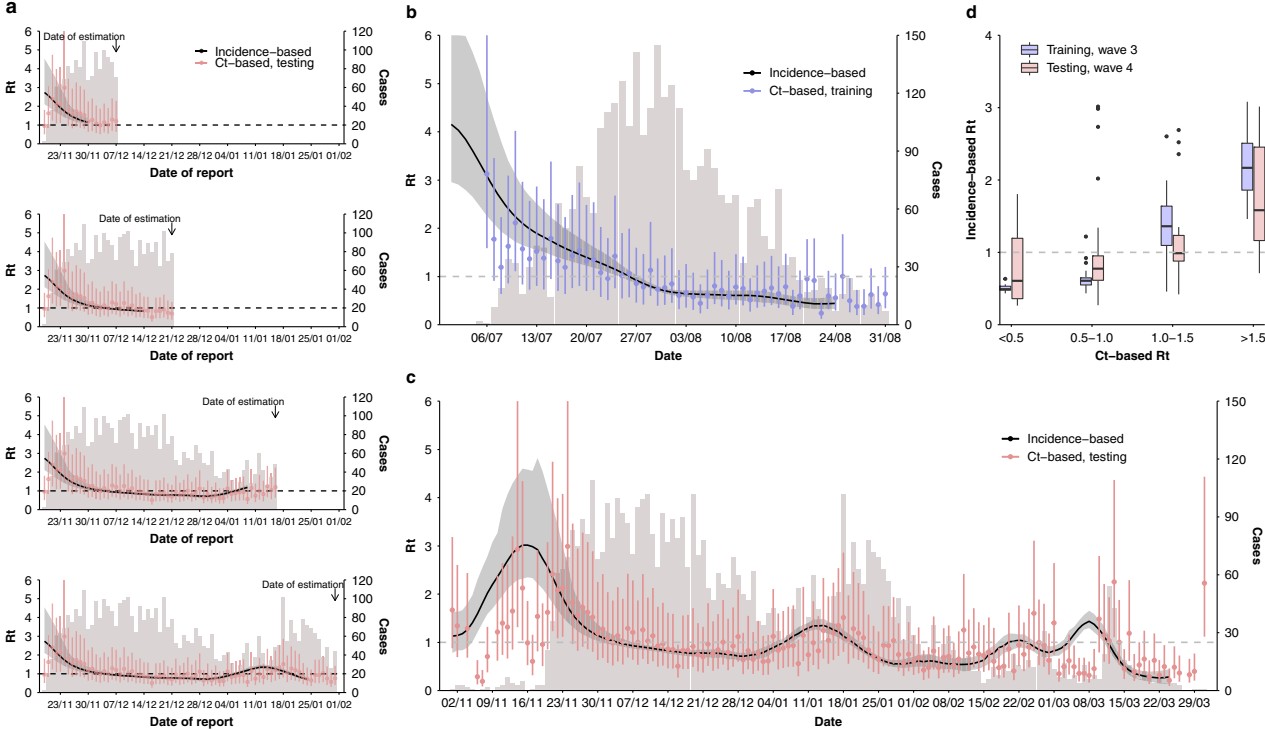

**Fig. 2 Nowcast of the transmission dynamics of COVID-19 using Ct distribution. a** Nowcasting $R_t$ using the Ct-based method over four representative weeks. Gray bars represent the number of laboratory-confirmed local cases by date of reporting, black lines and shaded areas indicate the mean and 95% CrIs for incidence-based $R_t$, while dots and vertical lines represent mean and 95% prediction intervals for Ct-based $R_t$ estimated from Eq. (7) (same for **b**, **c**). In this panel, n = 19, 33, 59, and 74 daily values from top to bottom panels sequentially. **b**, **c** Comparison of incidence-based $R_t$ and Ct-based $R_t$ over the training period (July 2020–August 2020, **b**) and the testing period (November 2020–March 2021, **c**). n = 57 daily values for training period in panel **b** and = 146 for testing period in panel **c** (same for **d**). **d** Distributions of incidence-based $R_t$ under various intervals of Ct-based $R_t$. Boxes represent the IQR and median of incidence-based $R_t$ under the corresponding interval of Ct-based $R_t$ over training (purple) and testing (pink) periods, lower whiskers represent the minimum and upper whiskers represent either the maximum or the largest values that are within the distance of 1.5 times the IQR of all incidence-based $R_t$ under various Ct- based $R_t$ intervals, dots represent values beyond the lower and upper whiskers.

real-time situational awareness using the Ct-based methods. In addition, we showed that the daily Ct distribution could be applied for tracking epidemics under a symptom and contact-tracing based setting, such as Hong Kong, providing empirical data to support the hypotheses generated from previous Ct-based studies[13]. Temporal changes in population Ct distribution over an epidemic largely reflect changes in infection-to-sampling delays that is determined by delays of infection-to-onset (i.e., incubation) and onset-to-testing (i.e., testing delay). We showed that the onset-to-testing did not provide additional information to the population Ct distribution (Supplementary Table 5), suggesting the observed changes in Ct distribution was largely driven by collective changes in the exposure time. In particular we demonstrated that epidemic progress might better explain temporal changes in population Ct distribution than changes in testing delays due to varied detection patterns in our case.

Our simplified Ct-based method also provides an approach for real-time estimation of $R_t$ without requiring intensive surveillance of COVID-19 (i.e., accurate daily case counts by onset or diagnosis), which is of great significance especially for areas and time periods with limited and/or changing surveillance capacity. As the main prerequisite of the model was the distribution of Ct values among confirmed cases, our findings were less sensitive to changes in case reporting (e.g., due to definition changes and/or testing capacity constraints), which, by contrast, could affect conventional incidence-based $R_t$ estimation if not accounted for[21,22]. For example, our results showed that population-level Ct distributions remained informative in tracking epidemic changes over time regardless of changes in surveillance in Hong Kong,

especially by expanding the testing capacity and therefore detecting more cases at earlier disease stage (i.e., asymptomatic cases; Supplementary Fig. 6) during the fourth wave in our case[15]. This was further supported by the high accuracy of Ct-based $R_t$ under various settings of case detection in our simulations. Of note, $R_t$ estimated with very few Ct samples (e.g., less than 30; Supplementary Table 4) collected on a given day can lead to larger uncertainty, though we believe this may not be an issue for most areas with prevalent local COVID-19 transmissions, even if testing capacity is limited.

Our work is not the first attempt to improve real-time tracking COVID-19 transmission. Another study[23] demonstrated that using sewage surveillance could shorten the prediction delays to two days ahead of test positives, which, however, did not fully overcome the delays between being infected and being PCR detectable[6]. In addition, the possibilities of locating sewage samples containing viral RNA could be low, especially when transmission in the community was low. Under such a situation, our method, which leverages existing information from confirmed cases, maybe less resource-consuming and faster to implement, as long as the reporting delay could be shortened.

Future applications of our method may need adaptation to different populations, especially among those with which viral load trajectories differ. In particular, populations with higher SARS-CoV-2 vaccination rates may expect increased average Ct values when $R_t$ is greater than 1, as lower viral loads were found in cases who had received COVID-19 vaccinations[24]. Similarly, increased Ct values when $R_t$ is greater than 1 may also be found in populations younger than Hong Kong's population, due to the

generally lower viral loads observed among young people[9]. As such, while we believe the intrinsic relationship between population viral loads and $R_t$ estimates will remain valid (as long as the time relation between infection and viral shedding still holds), recalibrations of the model may be needed when applying our model to different populations.

By the time this study was performed, there were limited VoCs circulating in Hong Kong[18], therefore we were not able to validate the generalizability of our model under outbreaks dominated by VoCs. A modeling study suggested that differences in population-level Ct values of samples from symptom-based surveillance were more likely to reflect changes in viral load trajectories instead of differences in transmission rates across strains[13,25]. As such, if increased viral loads (i.e., lower Ct values) occur with variant infections[26], this may lead to decreased average Ct values when $R_t$ is greater than 1. Therefore, monitoring the model performance and leveraging information on factors affecting viral load levels (e.g., genomic surveillance) are needed. For instance, unfavorable model performance (e.g., lower consistency between incidence-based and Ct-based estimates for more than a week) could indicate changes in correlations between population Ct distribution and epidemic progress. Under such cases, investigations about the driving factors of the divergence (e.g., changes in circulation strains or age-structure of the infected population) are needed to recalibrate the model.

To summarize, in this analysis we applied a simplified method to incorporate the population-level viral loads into the real-time estimation of transmission rates for COVID-19 under symptom-based surveillance. We demonstrated that the Ct-based method could provide accurate nowcasting of $R_t$ potentially allowing capacity-constrained regions to track local outbreaks quantitatively in a timely manner. Our method may need adaptions to different populations and the evolving strains, mainly to recalibrate the absolute extent to which the population viral loads correlate with COVID-19 transmission.

## Methods

**Study settings.** Hong Kong was among the first places to identify COVID-19 cases globally, with its first COVID-19 case detected in late January 2020[8,27]. Cases were classified as "imported cases", "local cases epidemiologically linked with imported cases", "unlinked local cases", and "local cases epidemiologically linked with local cases" according to their epidemiological characteristics and location of infection. All suspected COVID-19 cases were confirmed by RT-qPCR in a local centralized public health laboratory. A laboratory-confirmed COVID-19 case was defined as a local case if the case did not visit places outside of Hong Kong in the 14 days before symptom onset (for symptomatic cases) or confirmation (for asymptomatic cases); otherwise defined as imported cases.

By the time of analysis, four different waves of transmissions have occurred in Hong Kong. In this study, we restricted our analyses to the third (July 2020 to August 2020) and fourth (November 2020 to March 2021) waves which were dominated by local transmission, where 89% were local cases. Since we were interested in the local transmission of COVID-19, we only included local cases (unlinked local cases and local cases epidemiologically linked with local cases) in our analyses. Given the stringent border controls since July 2020[28] and the extremely small number of local cases linked with imported cases in Hong Kong (<0.1%), we assumed that all unlinked local cases were infected from other local cases. We did not include the first two waves (i.e., January to May 2020) as they were predominantly imported cases and smaller clusters linked to those imported cases[8,17].

In Hong Kong, local COVID-19 cases were generally detected from clinical diagnoses that targeted people with acute respiratory symptoms and from public health surveillance that targeted populations with predefined high risks of exposures (e.g., staff working at healthcare centers; residents living in neighborhoods with any lab-confirmed cases) by health authorities[15]. Upon case confirmation, contact tracing was carried out based on epidemiological information, with details described elsewhere[29]. Among 8646 local COVID-19 cases confirmed during our studied periods, 77% (6700 out of 8646) were detected symptomatic and 65% (5651 out of 8646) were found with epidemiological links with other known cases; across 23% (1946 out of 8646) of cases who were detected as asymptomatic, 81% (1584 out of 1946) of them were linked to other local cases. As such, the surveillance of COVID-19 in Hong Kong is largely symptom and contact-tracing-based.

## Data sources

*Data on viral load of COVID-19 cases.* In Hong Kong, all confirmed COVID-19 cases (including asymptomatic cases) were admitted to hospitals for isolation and standardized management, with their hospitalization records stored in the data system managed by Hospital Authority (HA). Results for SARS-CoV-2 RT-qPCR tests (LightMix® Modular SARS-CoV-2 (COVID-19) E-gene, TIB Molbiol/Roche, Berlin, Germany)[16] were recorded as Ct values in the system. The Ct value is the number of cycles needed to amplify the viral RNA in a specimen where the reported fluorescent signal reaches a pre-defined level in RT-qPCR assays. Therefore, the Ct value is inversely associated with viral load and could be used as a semi-quantitative measurement for viral load. In our main analysis, we used Ct values to measure viral load and analyzed the first recorded Ct value for each local case (which was usually sampled on or one day before admission) during the study period. Population viral load distributions were assessed by the date when samples were collected.

*Demographic and epidemiological information of confirmed COVID-19 cases.* We obtained demographic and epidemiological information from the Department of Health of the Government of Hong Kong, including age, date of symptom onset, and case classification (i.e., local, imported, and contacts of local or imported cases).

Ethical approval for this study was obtained from the Institutional Review Board of the University of Hong Kong (IRB No. UW 20-341).

## Statistical methods

*Estimating incidence-based $R_t$.* We estimated the incidence-based $R_t$ for local cases using an extension of Cori et al.[3,7,30]. Briefly, local COVID-19 cases confirmed on each day $t$, $Q_1(t)$ was used for deconvolution to estimate the number of infections on each day $t$, $Y_1(t)$[31]. We assumed an average 5.2 days (SD 3.9) for the incubation period[19] and an average 4.7 days (SD 3.2 days, unpublished data) delay between illness onset to reporting empirically observed in Hong Kong, which were used for deconvolution. In this framework, the daily local $R_t$ (i.e., the incidence-based $R_t$ in our analysis) was the ratio between the number of new local cases at time $t$, $Y_1(t)$, and the total infectiousness of cases at time $t$, given by $\sum_{k=1}^{t-1} Y_1(k)w_L(t-k)$, where $w_L(t-k)$ denote the probability of being infectious $t-k$ days after infections. The transmission was modeled by a Poisson process, and therefore, we have

$$Y_1(t) \sim Poisson\left\{ R_t \sum_{k=1}^{t-1} Y_1(k)w_L(t-k) \right\} \quad (1)$$

$w_L(t-k)$ was estimated using the convolution of the incubation period (mean 5.2 days, SD 3.9)[19] and the infectiousness relative to onset[5] (details described elsewhere[7]). To fully utilize available case count information and to provide more timely $R_t$ estimates under the incidence-based method, we used the smoothing method described in Cori et al.[30] and calculated $R_t$ estimates over a time window of size $\tau = 14$ ending at time $t$, assuming that the transmission rates was constant over the time period $[t-\tau+1, t]$. We used a Markov chain Monte Carlo algorithm to estimate the incidence-based $R_t$, and we assumed the prior for $R_t$ is Gamma (1,5) with mean and SD equal to 5[32]. To account for the uncertainty of other parameter such as the incubation period, we used an bootstrap approach in Salje et al[33] to reconstruct 200 epidemic curves and perform estimation. After that we presented the mean, 2.5% and 97.5% quantiles for those 200 $R_t$ estimates for each day $t$. More details about incidence-based $R_t$ estimation was described elsewhere[7].

*Temporal distribution of population-level Ct values.* We analyzed the first available Ct value record for each local COVID-19 case (i.e., $y_{j,t}$, $t$ is the calendar date when the first sample was collected for individual $j$). To characterize the temporal distribution of population-level Ct values over the study period, we fitted a generalized additive model (GAM) to the above-mentioned data over calendar time:

$$y_{j,t} = \alpha_0 + s(t) \quad (2)$$

where $s(t)$ was the smooth function for date $t$ over the study period. 95% confidence intervals (CIs) of the smoothed average daily Ct were derived from 500 bootstraps (Fig. 1b; Supplementary Fig. 1a). In each bootstrap, we resampled from the data on cases' first available Ct values and refitted the GAM. We also illustrated temporal changes in delays between illness onset to sampling and found a consistent pattern between the temporal trend of Ct distributions and that of delays (Supplementary Fig. 1). We did not include samples collected between 1 September 2020 to 31 October 2020 due to the small number of samples that were collected on each day.

To validate that the observed temporal variations in population-level Ct distribution was not driven by variations in individual viral load trajectories, we estimated the Ct value on the date of illness onset based on the observed pattern of Ct values against time-since-onset (Supplementary Fig. 2). We fitted a log-linear regression of the first available Ct value for individual $j$ ($y_j$) on the time interval between the individual's illness onset and first sample collection ($\delta_j$) and age group

($a_j$, modeled as categorical, i.e., 0–18, 19–64, and $\geq 65$ years old)

$$ln\left(y_j\right) = \beta_j + \beta_1 \delta_j + \beta_2 a_j + \beta_3 \delta_j a_j \tag{3}$$

where $\beta_1$, $\beta_2$, and $\beta_3$ are the estimated coefficients for the time interval between the illness onset and first sample collection, age group, and their interaction, respectively. We then calculated the back-projected Ct value at illness onset by setting $\delta_j = 0$. We chose the log-linear model as the Akaike information criterion (AIC) indicated it outperformed the linear model in terms of model fit (−9177 and 38,279 for the log-linear and linear models, respectively).

To compare differences in the temporal trend of Ct values at sampling and at onset, we fitted GAM of Ct values at sampling (Eq. (2)) (or at onset as in Eq. (4)) against the smoothed calendar time over the third wave when sample sizes were over 30 per day

$$\hat{y}_{j,t} = \hat{a}_0 + s(t) \tag{4}$$

where $\hat{y}_{j,t}$ is the extrapolated Ct value at illness onset for individual $j$ who had illness onset on the calendar date $t$. We calculated the mean and skewness of the Ct values at sampling or at onset over each bi-weekly window throughout the study period. Both results showed that Ct values at sampling were more variable than Ct values extrapolated at illness onset (Supplementary Figs. 3 and 4), suggesting variations in individual viral load trajectories may not be the major driver of the observed temporal variation in population-level Ct distribution over our study period during which only the wild-type SARS-CoV-2 strains have been circulating locally.

*Incorporating Ct distributions into $R_t$ estimation (Ct-based $R_t$).* We used the mean ($\bar{x}_t$) and skewness ($b_t$)[34] to measure the distribution of Ct values that were sampled on date $t$:

$$\bar{x}_t = \frac{1}{n_t} \sum_{i=1}^{n_t} y_{t,i} \tag{5}$$

$$b_t = \frac{\frac{1}{n_t} \sum_{i=1}^{n_t} (y_{t,i} - \bar{x}_t)^3}{\left[\frac{1}{n_t - 1} \sum_{i=1}^{n_t} (y_{t,i} - \bar{x}_t)^2\right]^{\frac{3}{2}}} \tag{6}$$

where $y_{t,i}$ represented the $i$th ($i = 1, 2, \ldots, n_t$) of the total $n_t$ Ct values that were sampled on day $t$. 95% CIs of the daily skewness $b_t$ were calculated from 500 bootstraps (Fig. 1c), with data on cases' first available Ct values resampled in each bootstrap to re-calculate the daily skewness.

We first calculated the Spearman's rank correlation coefficient ($\rho$) between daily Ct distribution (i.e., daily mean and skewness) and the natural log-transformed incidence-based $R_t$ (Supplementary Table 1). To determine the best fit model that characterized the association between daily Ct distribution and the incidence-based Rt, we compared AIC of a series of regression models over the training period (i.e., 6 July 2020 to 31 August 2020), which used different formats of dependent variable and measurements for predictive variables (Supplementary Table 3). We compared models that were fitted to linear scale and natural log-transformed incidence-based $R_t$. We also assessed models that included different combinations of measurements for daily Ct distributions, including mean, median, and skewness. We imputed the daily Ct distributions using the average of that within the preceding 7 days when no samples were collected on that day. The model fitted to natural log-transformed incidence-based $R_t$ ($\ln(R_t)$) on the daily mean ($\bar{x}_t$) and skewness ($b_t$) of Ct values was found with the lowest AIC and was used in our main analyses (Supplementary Table 3)

$$\ln(R_t) = \gamma_0 + \gamma_{\bar{x}} \bar{x}_t + \gamma_b b_t \tag{7}$$

where $\gamma_{\bar{x}}$ and $\gamma_b$ were coefficients for daily mean and skewness of Ct values from the regression model and were reported in Supplementary Table 2 after exponential transformation.

We explored the impact of the training period and sample sizes for our estimation. We trained our model over different training periods with various starting dates (either between 4 and 23 July 2020 or between 10 and 29 November 2020) and we set lengths of these alternative training periods like 30, 40, 50, and 60 days respectively, after which we compared their adjusted R square and demonstrated the time period covered by the best-fit model over training periods of the same length (Supplementary Fig. 8). For sample sizes, we calculated the Spearman correlation coefficients between incidence-based and Ct-based estimates under different sample size intervals and found that Ct-based $R_t$ tended to be more accurate with over 30 records per day (Supplementary Table 4).

To assess whether our results would be affected by the age distribution of cases who were sampled on each day, we performed a sensitivity analysis by including the mean age ($\bar{a}_t$) of cases whose first sample were collected on day $t$ into the above-mentioned main model (Eq. (7))

$$\ln(R_t) = \gamma_0 + \gamma_{\bar{x}} \bar{x}_t + \gamma_b b_t + \gamma_{\bar{a}} \bar{a}_t \tag{8}$$

Results suggested similar predictions from models with and without considering cases' age distribution (Supplementary Fig. 5).

To assess whether our results would be affected by changes in sampling strategies in Hong Kong, we first looked at temporal changes in the proportion of symptomatic cases among all confirmed local cases (Supplementary Fig. 6a). We performed a sensitivity analysis by fitting the main model (Eq. (7)) using only records from symptomatic cases and found no significant difference from our main results (Supplementary Fig. 6). We also adjusted for delays from illness onset to sampling in our main model (Eq. (7)) and found that changes in sample collections (coefficient $\beta = 0.93$, 95% CI: 0.87–1.01) did not alter the association between population Ct distribution and incidence-based $R_t$ (Supplementary Table 5).

*Cross-validations of the model.* To validate the generalizability of this Ct-based method, we fitted the main model (Eq. (7)) using data from an alternative training period, i.e., from 20 November 2020 to 19 December 2020 (the initial stage of the fourth wave) (Supplementary Fig. 7, Supplementary Table 2).

We further performed tenfold cross-validation by randomly splitting the data between 6 July 2020 and 31 March 2021 into ten validation sets, after excluding days with less than five available Ct samples were collected. For each validation, we held one set as a testing set and trained the remaining nine sets using the main model (Eq. (7)). We compared the consistency between the Ct- and incidence-based $R_t$ for the testing set by calculating the proportion of days when the two estimates were simultaneously below or above 1 (i.e., in the same direction) over the total duration of each validation set. We also assessed the prediction performance using the mean absolute error (MAE) for the Ct- ($E(R_t)$) and incidence-based $R_t$ for each validation set

$$MAE = \frac{\sum_{d \in D} \left|\ln(E(R_d)) - \ln(R_d)\right|}{N_D} \tag{9}$$

where $d$ is a given date in the validation set $D$ and $N_D$ is the number of days included in each validation set. We found an average of 0.28 (ranging from 0.25 to 0.34) of the MAE across ten validation sets, suggesting a good performance of our model predictions.

## Simulations

*Transmission model.* We used a susceptible-exposed-infectious-recovered (SEIR) model to simulate two consecutive epidemic waves assuming a closed population ($n = 7.5$ million, approximately the same size to the population in Hong Kong) and initial infections of 0.001%. Briefly, we simulated infections with a stochastic SEIR model, with compartments for susceptible (S), exposed-but-not-yet-infectious (E), infectious (I), and recovered (R). The compartmental transition equations are listed below:

$$\frac{dS}{dt} = \frac{-\beta_t S(t) I(t)}{N}$$
$$\frac{dE}{dt} = \frac{\beta_t S(t) I(t)}{N} - \sigma E(t)$$
$$\frac{dI}{dt} = \sigma E(t) - \gamma I(t) \tag{10}$$
$$\frac{dR}{dt} = \gamma I(t)$$

where $\beta_t = \frac{R_0}{\gamma}$ for $t \geq t_0$. $1/\sigma$ ($\sigma = 5 days$[19]) indicated the average time for individuals to transit from E to I, while $1/\gamma$ ($\gamma = 4 days$[13]) referred to the observed mean infectious period. Detailed descriptions of parameters were listed in Supplementary Table 6.

We used synthetic $\beta$ (which determines the underlying transmission rate) that changes over time $t$ to synthesize the process of two consecutive epidemic waves

$$\beta_t = \begin{cases} R_0 \gamma & t < 60 \\ R_0^2 \gamma & 60 \leq t < 110 \\ R_0^3 \gamma & 110 \leq t < 150 \\ R_0^4 \gamma & t \geq 150 \end{cases} \tag{11}$$

where $R_0 = 2.2$, $R_0^3 = 1.9$ and $R_0^2 = R_0^4 = 0.3$. Epidemic switch points were set at day 60 and 110 and $R_0$ changes between switch points were fitted via a cubic smoothing spline and interpolated into smooth transitions. $R_t$ calculated under this SEIR model (denoted as the simulation truth) would then be

$$R_t = \frac{S(t)}{N} \beta_t \gamma \tag{12}$$

*Symptom-based case detections.* In symptom-based surveillance, we assumed that only individuals who developed symptoms after infections (which follow a binomial distribution with $p_{Sym|Inf} = 0.6$[35,36]) would be detected after illness onset. We assumed the incubation period followed log-normal distribution (mean = 5.2, SD = 3.9)[19], while we estimated the delays between onset to detection with a gamma distribution (shape = 1.83 and rate = 0.43) using observations from Hong Kong.

We simulated four different scenarios to represent various intensities of case detection (Supplementary Fig. 9):

1. Scenario 1: a fixed detection probability of 25%. We used this scenario to represent the practice of stable detection, as the case in Hong Kong[15].

2. Scenario 2: a fixed detection probability of 10%. We used this scenario to represent the situation of stable but limited detection capacity.

3. Scenario 3: the probability of detection increased from 15% to 60% over the second simulated wave. We used this scenario to represent the situation of expansion in case definition, as in the initial stage of the outbreak in mainland China[22].

4. Scenario 4: a fixed detection probability of 25% except for the under-detection (lowest at 5%) during the initial stage of the second simulated wave.

*Individual viral load trajectory.* We simulated the viral load trajectories over the infection course for each detected symptomatic case using the previously published method[37]. We assumed a unimodal trend of Ct value changes that will reach the lowest on the date of illness onset (and therefore had the same distribution of incubation periods[19]), with the lowest Ct value (i.e., peak viral load) following a normal distribution with a mean of 22.3 and SD of 4.2[11]. The duration of viral shedding since onset was parameterized as normally distributed with mean and SD of 17 and 0.94 days[38]. Each infected individual, if detected, would then have their corresponding sampled Ct values as the Ct value falling on day $k$ post infection based on their own Ct trajectories, with $k$ being the time interval between their dates of infection and detection.

*Daily $R_t$ and population-level Ct from simulations.* Incidence-based $R_t$ from synthetic case count data was estimated using the R package EpiNow2[3] that has accounted for delays and other sources of uncertainty in a more sophisticated way. The incubation period and reporting delay that were used for deconvolution were assumed to follow the delay distributions that were simulated from symptom-based surveillance. The mean and variance of the generation interval under the SEIR model were specified as $T_c = 1/\sigma + 1/\gamma$ and $Var = 2(\frac{T_c}{2})^2$ respectively, with $\sigma$ and $\gamma$ being the average time for individuals to transit from E to I and from I to loss of infectiousness respectively (see Supplementary Table 6). More details were provided in https://github.com/epiforecasts/EpiNow2.

The daily distribution of population Ct from the simulations was estimated by mean and skewness, as in Eqs. (5) and (6). The regression model (Eq. (7)) used to generate Ct-based $R_t$ under each scenario was selected by comparing the adjusted $R$ square of models fitted over different training periods during the first simulated wave (Supplementary Fig. 10), after which we applied the model to estimate the Ct-based $R_t$ for days following the training period (denoted as the testing period). Spearman correlation coefficients $\rho$ between Ct-based $R_t$ and the simulation truth were calculated to evaluate the accuracy of our estimates.

To investigate the uncertainty in sampling Ct values and therefore the accuracy of Ct-based $R_t$ estimates, we repeated each scenario 100 times and calculated the Spearman correlation coefficient between estimated Ct-based $R_t$ and the simulation truth for each simulation. We calculated the mean, 2.5 and 97.5% quantiles of the correlation coefficient across 100 simulations for each scenario (Supplementary Table 7).

All statistical analyses were conducted in R version 4.1.2 (R Development Core Team, 2021).

**Reporting summary**. Further information on research design is available in the Nature Research Reporting Summary linked to this article.

## Data availability

All demographic and epidemiological information of confirmed COVID-19 cases is freely available from the Centre for Health Protection website (https://www.coronavirus.gov.hk/eng/index.html). Daily aggregate data (including case counts, incidence-based $R_t$ and Ct distributions), and simulation data generated in this study have been deposited in the GitHub repository.

## Code availability

All codes for analyses are available at the GitHub repository.

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

## Acknowledgements

We thank the Department of Health and Hospital Authority of the Food and Health Bureau of the Government of Hong Kong for providing the data for the analysis. This project was supported by the Health and Medical Research Fund, Food and Health Bureau, Government of the Hong Kong Special Administrative Region (grant no. COVID190118; B.J.C.), and the Theme-based Research Scheme (Project No. T11-712/19-N; B.J.C.) of the Research Grants Council of the Hong Kong SAR Government.

## Author contributions

All authors meet the ICMJE criteria for authorship. The study was conceived by B.J.C. and B.Y. Y.L., E.H.Y.L., J.Y.W., H.S.B., J.K.C., F.H., H.G., and T.K.T. prepared the data. B.Y. and Y.L. developed the model. Y.L. and B.Y. conducted the data analyses. S.C., D.C.A., S.T.A., N.H.L.L., T.K.T., P.W., G.M.L., and B.J.C. interpreted the results. Y.L. and B.Y. wrote the first draft of the paper. All authors provided critical review and revision of the text and approved the final version.

## Competing interests

B.J.C. consults for AstraZeneca, GSK, Moderna, Roche, Sanofi Pasteur, and Pfizer. The remaining authors declare no competing interests.
