## [Peer Review File · Nature Communications]

Incorporating temporal distribution of population-level viral load enables real-time estimation of COVID-19 transmissionEditorial Note: Parts of this Peer Review File have been redacted as indicated to remove third-party material where no permission to publish could be obtained.

REVIEWER COMMENTS

Reviewer #1 (Remarks to the Author):

This is an interesting study that aims to provide an alternative method using population-level viral load to nowcast COVID-19 transmissibility. By analyzing viral load data in Hong Kong, the authors identified a correlation between the distribution of Ct values and the Rt estimated based on case counts. They further proposed a regression model to estimate Rt using the mean and skewness of Ct distributions. The study addresses a practical problem in infectious disease surveillance, and the proposed method could be potentially applied in other locales in real time. Below I have a few questions on the analyses and the real-world application.

1. The authors examined the Spearman's rank correlation between incidence-based Rt and average Ct value. How about the Pearson's correlation? If the goal is to estimate Rt values, maybe the Pearson's correlation is more relevant?
2. In line 173, the authors provide an example on the limitation of incidence-based Rt (sensitive to case reporting change). In the main text, the Ct-based Rt is validated using incidence-based Rt. It seems a bit circular to use an imperfect "ground truth" to validate the method. Could authors show some evidence that such occasion is rare? For instance, testing data to show case reporting is stable over the majority of the study period?
3. "The key element of a suitable training period is to cover a period with sufficient samples to reflect epidemic changes, which usually starts from 2-3 weeks before the case peak till around 1-2 weeks after the peak." Does the training period have to cover a peak? In some regions (like in US), there might not be a clear peak. In addition, if two successive peaks are driven by different strains with potentially different viral dynamics, the generalization of the model from one peak to another may be a problem. A discussion on this point would be helpful.
4. The uncertainty of Ct-based Rt seems high in Fig. 2 – most of the 95% CIs cover $R_t=1$, an important threshold in public health intervention. Should decision-makers use the mean estimate?
5. It might be good to provide some practical guidance for generalization to other locations. For instance, how often to recalibrate the model? What's the criterion to recalibrate? How many Ct samples are enough to get meaningful and robust Rt estimation?

Sen Pei

Reviewer #2 (Remarks to the Author):

Lin and Yang et al. describe a method for estimating the effective reproductive number, R_t , using a novel data stream: RT-qPCR cycle threshold (Ct) values from routine case finding. This work confirms and builds on the recently described link between the population-level distribution of Ct values and the incidence of SARS-CoV-2 infection. Although the significance of these findings is fairly limited, a strength of the proposed method is that it is simple to implement compared to the other Ct-based method in the literature. The work is timely and well written, adding original research to an underexplored topic in infectious disease dynamics. The methodology appears to be appropriate, reproducible, and accompanied by clear code.

My main concern is that it is not made clear how, despite the claim, the Ct-based method improves upon the current R_t -estimation paradigm using case counts. The same data are used for case- and Ct-based R_t estimation, and because the ground truth is not known, it is not clear which method is more reliable. If sufficient data are available to estimate R_t using case counts (and are needed to calibrate the Ct model), what is the value of using the Ct values? The added value of using Ct values from samples collected in this way in capacity-constrained regions could be demonstrated in simulation and should at least be explained more clearly. Specific comments below.

Major comments

1. Confounding from changing sample collection

The authors note in the methods that the samples are largely from symptomatic cases or those with epidemiological links to other local cases. I would suggest investigating if these proportions changed over time and considering what impact their changing proportions would have on the observed Ct trends. One question that comes to mind is if the Ct distributions varied by source of sample. For example, if the proportion of samples from symptomatic individuals increased, then we might expect Ct values to become lower on average irrespective of changes in incidence. An extreme consequence might be that changes in Ct do explain variation in Rt, but simply due to systematic changes in who is getting tested rather than representation of old vs. new infections. Overall, the impact of varying sampling strategies/make up on the observed trends should be explored discussed.

2. Sensitivity analysis of back-projected Ct values

Back-projecting the Ct values to time of illness onset is a cool idea, and I would encourage the authors to expand on this analysis. Specifically:

1. Please report the regression coefficients and CIs. The coefficient for time-since-onset gives an estimate of viral clearance rate and can be compared to the existing literature. A plot of these data with model fits would also be helpful.
2. What do the distribution of delays between onset and sample collection look like over time? Suggest a plot similar to Fig S1. As the authors know, we would expect changes in the time-since-onset distribution to drive the changes in Cts, which is where the information for estimating Rt using Cts comes from. I don't think this is currently made clear to the reader.
3. Related, I'd suggest delving a bit more into the comparison of the Ct-at-onset distributions over time. We would not expect the distribution of Cts *at the time of onset* to change with calendar time, as supported by the coefficient of variation analysis. However, it is difficult to make this visual comparison using the current Fig S2, so some text description and perhaps statistical tests to investigate whether the Ct-at-onset distribution changes over time would be helpful.
4. Is there a reason that only skewness is compared and not the median/mean Ct?

3. Added value of Ct-based methods

It is not particularly obvious how the Ct-based method mitigates the right-censoring issue of incidence-based Rt estimation, other than the fact that the chosen incidence-based Rt estimates are truncated 7 days earlier to avoid dealing with right censoring, whereas the Ct-based method includes data up to the current day. However, incidence-based Rt methods can deal with right-censoring by including a nowcasting/forecasting component (eg. Abbott & Hellewell et al. Wellcome Open Research 2021 <https://doi.org/10.12688/wellcomeopenres.16006.2>), so it is not quite a fair comparison. It also looks like the incidence-based method panels in Fig 2 are from fitting to the entire dataset and then sub setting by the relevant time windows, rather than re-fitting the Rt model using data that *would* have been available on a given date. If the authors stand by the claim that the Ct method is not affected by right censoring but the incidence-based method is, then I encourage the authors to explain the intuition and to consider if the comparison shown here is fair.

It is also not clear how these findings support the claim on L163 that "Our simplified Ct-based method also provides an approach for real-time estimation of Rt without requiring intensive surveillance of COVID-19". Neither sample size requirements nor simulation settings with limited data are explored here. To support this claim, I suggest that the authors discuss sample sizes and sample collection frequency much more explicitly -- how many samples are required for robust incidence-based Rt estimation versus Ct-based estimation? Support for these claims with simulation studies would be helpful (eg. simulate incidence curves and Ct values and compare estimates using incidence-based and Ct-based Rt estimation with different sample sizes).

Minor comments

- Title: I might replace "transmissibility" with "transmission rates" or just "transmission". To me, transmissibility is a fundamental property of the virus, not of the population process.
- Abstract: the term RT-qPCR does not appear in the abstract, which is crucial to understand what you are referring to with the term "cycle threshold values". Eg. L35 "measured by RT-qPCR cycle threshold values".

- Introduction: I suggest adding a few additional references on R_t estimation, as the current reference list does not reflect the state-of-the-art. For example: Abbott & Hellewell et al. Wellcome Open Research 2021 <https://doi.org/10.12688/wellcomeopenres.16006.2>; Flaxman et al. Nature 2020 <https://doi.org/10.1038/s41586-020-2405-7>. Also a very recent (since the authors submitted this piece) paper by Parag PLOS Comp Biol 2021 <https://doi.org/10.1371/journal.pcbi.1009347>.
- L57: suggest an additional reference Kissler et al. PLOS Biology <https://doi.org/10.1371/journal.pbio.3001333>
- Introduction: the aims of the paper feel quite weak at the moment, stating only that the aim is to “determine whether inclusion of data on C_t values could allow real-time estimation of transmissibility”. This has already been shown to be possible, and so I urge the authors to elaborate on what this paper adds, eg. using C_t values from symptom-based surveillance rather than random cross-sectional samples as shown previously; allowing R_t to be estimated with less delay than using case counts alone.
- A key condition in Hay & Kennedy-Shaffer et al. was the use of random cross-sectional samples. It is less clear how samples are obtained here in the main text, though I note that this is explained nicely in the methods. I would suggest i) moving the details on the sample makeup (ie. proportion of symptomatics etc) to the Results section, and ii) adding to the discussion how sample collection strategy impacts how C_t values can be used.
- What was the gene target used to generate the C_t values?
- Fig 2: make clearer in panels a-d which data are available in each window (eg. add a vertical dashed line or truncate the case counts).
- L147: what is the basis for calling these estimates precise? Many of the confidence intervals seem far wider than the incidence-based R_t estimates.
- L154: I would argue that the cited methods provide longitudinal growth rate estimates, which are comparable to R_t given that growth rate and reproductive number can be linked using knowledge of the natural history of SARS-CoV-2.
- Fig 2 and S1: it would be useful to have some indication of sample size for in each of these time windows. For example, are the very differing R_t estimates in 2c driven by very small C_t sample sizes?
- L187 typo “chances”.
- L381: typo “rest nine sets”

Reviewer #1 (Remarks to the Author):

Reviewer1Comm1: This is an interesting study that aims to provide an alternative method using population-level viral load to nowcast COVID-19 transmissibility. By analyzing viral load data in Hong Kong, the authors identified a correlation between the distribution of Ct values and the Rt estimated based on case counts. They further proposed a regression model to estimate Rt using the mean and skewness of Ct distributions. The study addresses a practical problem in infectious disease surveillance, and the proposed method could be potentially applied in other locales in real time. Below I have a few questions on the analyses and the real-world application.

Response: Thank you for your supportive comment. We have provided our point-by-point responses below.

Reviewer1Comm2: The authors examined the Spearman’s rank correlation between incidence-based Rt and average Ct value. How about the Pearson’s correlation? If the goal is to estimate Rt values, maybe the Pearson’s correlation is more relevant?

Response: Thank you for your comment. Using the Pearson’s correlation lead to similar results with that from Spearman’s rank correlation (table below). We used Spearman’s rank correlation because the Ct value distribution is often skewed and therefore does not follow a normal distribution, which is however a requirement for Pearson’s correlation.

	Wave 3		Wave 4	
	coefficient	p-value	coefficient	p-value
Mean Ct				
Spearman’s rank correlation ρ	-0.79	<0.001	-0.52	<0.001
Pearson’s correlation β	-0.86	<0.001	-0.52	<0.001
Ct skewness				
Spearman’s rank correlation ρ	0.80	<0.001	0.27	<0.001
Pearson’s correlation β	0.78	<0.001	0.27	<0.001

Reviewer1Comm3: In line 173, the authors provide an example on the limitation of incidence-based Rt (sensitive to case reporting change). In the main text, the Ct-based Rt is validated using incidence-based Rt. It seems a bit circular to use an imperfect “ground truth” to validate the method. Could authors show some evidence that such occasion is rare? For instance, testing data to show case reporting is stable over the

majority of the study period?

Response: Thank you for your comment. We used the incidence-based R_t to train our model as we believe that the “ground truth” is not available most of the time but can be largely captured by incidence-based R_t , especially when not in the initial stage of the outbreak^{1,2}. We further conducted a series of simulation analyses, which also showed that the incidence-based R_t remained stably consistent with the simulation truth under most circumstance (e.g. scenarios 1-3), except for time periods with sudden abrupt changes (e.g. under detection as in the second wave in scenario 4). More importantly, simulation results suggested that the trained Ct-based R_t could recover the simulation truth under scenarios with limited and changing detection (Supplementary Fig. 10).

In Hong Kong, proportions of symptomatic cases remained high during our studied period (averagely 77%; Supplementary Fig. 6a). We also used Ct values from all confirmed cases and from symptomatic cases only to repeat the analyses, and found no significant difference in the Ct distributions and the Ct-based R_t , indicating comparatively stable predictions using either source of data (Supplementary Fig. 6b-d and in lines 135-138):

“We conducted sensitivity analyses to account for the potential impact of age on Ct distributions (Supplementary Fig. 5) and for changes in proportions of symptomatic cases (Supplementary Fig. 6) and the resulting Ct-based R_t , and found that the high correlation between Ct- and incidence-based R_t remained.”

Reviewer1Comm4.1: “The key element of a suitable training period is to cover a period with sufficient samples to reflect epidemic changes, which usually starts from 2-3 weeks before the case peak till around 1-2 weeks after the peak.” Does the training period have to cover a peak? In some regions (like in US), there might not be a clear peak.

Response: Thank you for your comment. We agree that it is important to make our proposed method applicable to places without clear epidemic peaks, like the US. Our analysis on the selection of training period showed that the regression model fit well as long as the training period covered the transition period of both the epidemic (i.e., from increase to decrease) and the population Ct, which usually covers the peak in places with clear epidemic waves (Supplementary Fig. 8). Furthermore, our sensitivity

analyses (Supplementary Fig. 7) and simulations (Supplementary Fig. 10) both suggested that the training period does not necessarily have to cover a complete wave/peak, while it needs to cover the transition period when R_t shift around 1. In particular, our sensitivity analysis using the earlier half of the fourth wave (i.e., a wave without a clear peak in Hong Kong) for training provided similar results to that from our main regression model (Supplementary Fig. 7a).

To address your comment, we have revised the text in lines 148-154 to make the selection of training period more generalizable to places without clear peaks.

“In addition, we found that our model predictions were insensitive to the selection of the training period as long as the training period had sufficient samples (e.g., > 30 samples per day as suggested in Supplementary Table 4) and could reflect changes in both epidemic growth and population Ct distributions. Such training period often covers the transition point when R_t shift around 1 and would span an epidemic peak in places with clear waves (Supplementary Fig. 8).”

Reviewer1Comm4.2: In addition, if two successive peaks are driven by different strains with potentially different viral dynamics, the generalization of the model from one peak to another may be a problem. A discussion on this point would be helpful.

Response: Thank you for the comment. We agree with you that our methods may need recalibrations before being generalized to waves that are caused by new variants or strains. Unfortunately, we were not able to test this with empirical data, as there were no local outbreaks caused by the emerging variants due to the stringent control measures in Hong Kong.

To address your comments, we discussed the potential challenges when generalizing our methods to new waves caused by different strains and suggested continuously monitoring epidemic situations by checking consistency between incidence-based R_t and population Ct distribution and by integrating genomic surveillance data (i.e., using genomic sequencing data to monitor potential changes in circulating pathogens). By doing so would inform us potential changes in correlations between R_t and population Ct and the possible timing of recalibration of our applied models. We added the following text in lines 240-246:

“Therefore, monitoring the model performance and leveraging information on factors affecting viral load levels (e.g., genomic surveillance) are needed. For instance, unfavorable model performance (e.g., lower consistency between incidence-based and Ct-based estimates for more than a week) could indicate changes in correlations between population Ct distribution and epidemic progress. Under such cases, investigations about the driving factors of the divergence (e.g., changes in circulation strains or age-structure of infected population) are needed to recalibrate the model.”

Reviewer1Comm5: The uncertainty of Ct-based R_t seems high in Fig. 2 – most of the 95% CIs cover $R_t=1$, an important threshold in public health intervention. Should decision-makers use the mean estimate?

Response: Thank you for your comment. We hope to clarify that several analytic factors would contribute to the wider interval of the Ct-based predictions. Firstly, as we have used a simple regression model to generate the Ct-based R_t , the interval shown for the Ct-based estimate was prediction interval, which is usually wider than the 95% credible interval that was reported by the incidence-based method. This is because the prediction interval is derived from simulations of a small sample size. Therefore, the wider prediction interval could be alleviated with larger sample sizes, as suggested by our simulation result (Supplementary Fig. 10).

In addition, differences in the length of smoothing window could also contribute to the width of the uncertainty intervals. We used the daily distribution to estimate the Ct-based R_t , meaning that we used a 1-day smoothing window for our estimation. Whereas, the incidence-based methods used a 14-day smoothing window, which is expected to give a narrower uncertain intervals of estimates than that from a 1-day smoothing window¹.

Given the above, we believe that point estimates from the Ct-based method could be more informative, which was also supported by the high consistency between the predictions and observations for both training (Spearman’s correlation coefficient, $\rho = 0.81$, $P < 0.001$) and testing periods ($\rho = 0.48$, $P < 0.001$).

Reviewer1Comm6: It might be good to provide some practical guidance for

generalization to other locations. For instance, how often to recalibrate the model? What's the criterion to recalibrate? How many Ct samples are enough to get meaningful and robust R_t estimation?

Response: Thank you for your suggestion. We believe that real-time monitoring the consistency between Ct-based and incidence-based R_t could help to determine the frequency and criteria for recalibration. For example, if the Spearman correlation coefficient between Ct-based and incidence-based R_t drops below 0.4 for more than a week, this could be a sign for recalibrating the model for future use. We think using additional sources of information could also help to monitor the model performance and to inform the timing of recalibration. For example, genomic surveillance could inform changes in circulating strains and indicate a need to recalibrate the model. To address your comment, we have revised our text in discussion (lines 240-246) as follows:

“Therefore, monitoring the model performance and leveraging information on factors affecting viral load levels (e.g., genomic surveillance) are needed. For instance, unfavorable model performance (e.g., lower consistency between incidence-based and Ct-based estimates for more than a week) could indicate changes in correlations between population Ct distribution and epidemic progress. Under such cases, investigations about the driving factors of the divergence (e.g., changes in circulation strains or age-structure of infected population) are needed to recalibrate the model.”

Regarding to the suitable sample size, we compared the consistency between Ct-based and incidence-based R_t under different sample sizes, and found that > 30 samples a day would be more sufficient to provide robust Ct-based R_t estimation. The correlations between the two methods are often higher than 0.5 when there were more than 30 samples a day (Supplementary Table 4). To address this comment, we have added the following in lines 413-417:

“For sample sizes, we calculated the Spearman correlation coefficients between incidence-based and Ct-based estimates under different sample size intervals and found that Ct-based R_t tended to be more accurate with over 30 records per day (Supplementary Table 4).”

Reviewer #2 (Remarks to the Author):

Reviewer2Comm1: Lin and Yang et al. describe a method for estimating the effective reproductive number, R_t , using a novel data stream: RT-qPCR cycle threshold (Ct) values from routine case finding. This work confirms and builds on the recently described link between the population-level distribution of Ct values and the incidence of SARS-CoV-2 infection. Although the significance of these findings is fairly limited, a strength of the proposed method is that it is simple to implement compared to the other Ct-based method in the literature. The work is timely and well written, adding original research to an underexplored topic in infectious disease dynamics. The methodology appears to be appropriate, reproducible, and accompanied by clear code.

My main concern is that it is not made clear how, despite the claim, the Ct-based method improves upon the current R_t -estimation paradigm using case counts. The same data are used for case- and Ct-based R_t estimation, and because the ground truth is not known, it is not clear which method is more reliable. If sufficient data are available to estimate R_t using case counts (and are needed to calibrate the Ct model), what is the value of using the Ct values? The added value of using Ct values from samples collected in this way in capacity-constrained regions could be demonstrated in simulation and should at least be explained more clearly. Specific comments below.

Response: Thank you for your comments and suggestions. We hope to clarify that, other than improving the accuracy of current incidence-based R_t estimation, we aimed to improve the timeliness of R_t estimation using the Ct-based method. Particularly, we aimed to incorporate Ct values into real-time predictions of R_t , by constructing a link between population Ct distribution and incidence-based R_t . We agree with you that the “ground truth” of R_t is often unknown, while previous study suggested that Cori’s method is likely to recover the true values of R_t after excluding the start of the outbreak^{1,2} (which was the case for our training period). The added value is that our proposed method could provide real-time prediction using the temporal information from the Ct distribution of daily samples, whereas the incidence-based methods are subject to the unavoidable delay that is caused by the latent period³. In particular, Fig. 2a indicated that Ct-based methods can provide precise estimation during the truncated week for incidence-based R_t .

Aside from enabling R_t nowcasting, we found that our Ct-based method is less sensitive to changes in case reporting patterns. We performed simulations under different scenarios of case detection and evaluated the performance of predictions of Ct-based R_t compared to the simulation truth (Supplementary Fig. 10). We found that, even under limited or changing case detection (scenarios 2-4), the accuracy of Ct-predicted R_t remained high (Supplementary Fig. 10; Supplementary Table 6).

Major comments

Reviewer2Comm2: ## 1. Confounding from changing sample collection

The authors note in the methods that the samples are largely from symptomatic cases or those with epidemiological links to other local cases. I would suggest investigating if these proportions changed over time and considering what impact their changing proportions would have on the observed Ct trends. One question that comes to mind is if the Ct distributions varied by source of sample. For example, if the proportion of samples from symptomatic individuals increased, then we might expect Ct values to become lower on average irrespective of changes in incidence. An extreme consequence might be that changes in Ct do explain variation in R_t , but simply due to systematic changes in who is getting tested rather than representation of old vs. new infections. Overall, the impact of varying sampling strategies/make up on the observed trends should be explored discussed.

Response: Thank you for your comment. As suggested, we investigated the temporal distribution of proportions of symptomatic cases and its potential impact on the Ct-based R_t estimation. We found that symptomatic cases constitute the majority of confirmed cases over the study period with an overall proportion of 77%, and had no significant impact on the Ct-based R_t estimates (Supplementary Fig. 6). To address your comment, we added the following in lines 135-138:

“We conducted sensitivity analyses to account for the potential impact of age on Ct distributions (Supplementary Fig. 5) and for changes in proportions of symptomatic cases (Supplementary Fig. 6) and the resulting Ct-based R_t , and found that the high correlation between Ct- and incidence-based R_t remained.”

Reviewer2Comm3: ## 2. Sensitivity analysis of back-projected Ct values ##

Back-projecting the Ct values to time of illness onset is a cool idea, and I would encourage the authors to expand on this analysis. Specifically:

Reviewer2Comm3.1: Please report the regression coefficients and CIs. The coefficient for time-since-onset gives an estimate of viral clearance rate and can be compared to the existing literature. A plot of these data with model fits would also be helpful.

Response: Thank you for your suggestion. Our estimated daily increase in Ct values was 1.057 (95% confidence interval (CI): 1.050, 1.063) (Supplementary Fig. 2), which was consistent with previous estimates^{4,5}. We have added in the estimated coefficient for time-since-onset in the results (in lines 114-118):

“To confirm that the changes in the observed daily Ct distribution were mostly driven by the epidemic dynamics despite individual variations in viral shedding, we extrapolated the Ct value back to illness onset for symptomatic cases using the fitted association that Ct values would increase 1.057 (95% confidence interval (CI): 1.050, 1.063) per day after illness onset (Supplementary Fig. 2; see Methods).”

We also plotted the raw data and the fitted model in Supplementary Fig. 2, which showed that Ct values increased with time since illness onset (i.e., viral loads decline) and that Ct values among the elderly tended to increase more slowly compared to the younger age groups.

Reviewer2Comm3.2: What do the distribution of delays between onset and sample collection look like over time? Suggest a plot similar to Fig S1. As the authors know, we would expect changes in the time-since-onset distribution to drive the changes in Cts, which is where the information for estimating Rt using Cts comes from. I don't think this is currently made clear to the reader.

Response: Thank you for your suggestion. We provided the temporal distribution of testing delay in Supplementary Fig. 1b. Temporal variations in testing delays ranged around 1-5 days (median: 3 days), with a slight delay during the epidemic peaks, which was likely due to the impact of epidemic growth (as indicated in Figure S4 in Hay's study⁶).

In a symptom-based surveillance system, changes in population Ct distribution can be affected by both the epidemic progress (i.e., distribution of exposure time) and detection pattern (i.e., time since onset to sample collection⁶). We further adjusted for the time from symptom onset to sample collection in our main model (Eq. 7) and found that changes in sample collections (coefficient $\beta = 0.93$, 95% CI: 0.87-1.01) did not alter the association between population Ct distribution and incidence-based R_t . Therefore, we think the epidemic progress is more likely to explain the temporal changes in the observed population Ct distribution than changes in reporting delays in our case.

To address your comment, we have revised Supplementary Fig. 1 to show distribution of delays between onset and sample collection and added the following in lines 191-193: *“In particular, we demonstrated that epidemic progress might better explain temporal changes in population Ct distribution than changes in testing delays due to varied detection patterns in our case.”*

Reviewer2Comm3.3: Related, I'd suggest delving a bit more into the comparison of the Ct-at-onset distributions over time. We would not expect the distribution of Cts *at the time of onset* to change with calendar time, as supported by the coefficient of variation analysis. However, it is difficult to make this visual comparison using the current Fig S2, so some text description and perhaps statistical tests to investigate whether the Ct-at-onset distribution changes over time would be helpful.

Response: Thank you for your comment and suggestions. We added Supplementary Fig. 3 to compare temporal distributions of Ct values at sampling and at onset during the third wave. By restricting to a time period with sufficient samples (i.e., >30 records per day), the observed changes in population Ct trend is more likely to be attributed to epidemic progress rather than individual variations with small sample size^{7,8}. Supplementary Fig. 3a shows that the smoothed Ct at onset was less variable (i.e., mostly centre around the mean) whereas the spline of Ct at sampling declined significantly during the compared time period.

To address this comment, we revised the text as below (in lines 118-122):

“We found that distributions of Ct values at onset were less variable than those at sampling during the studied period (coefficient of variation for skewness: 0.37 vs. 0.80)

(Supplementary Figs. 3 and 4), suggesting a relatively stable peaking level of viral loads across individuals over the course of the epidemic.”

Reviewer2Comm3.4: Is there a reason that only skewness is compared and not the median/mean Ct?

Response: Thank you for your comment. We originally presented the comparison of the skewness hoping to show that distribution of Ct by sampling was more variable compared to the distribution of Ct by onset. However, we agree with you that mean Ct may also help to reflect this point, as suggested by our main results (Fig. 1). To address your comment, we indicated the mean value of Ct values during compared time periods in Supplementary Fig. 4 to facilitate the comparisons for both skewness and mean distribution of Ct values over epidemic waves.

Reviewer2Comm4: ## 3. Added value of Ct-based methods ##

Reviewer2Comm4.1: It is not particularly obvious how the Ct-based method mitigates the right-censoring issue of incidence-based Rt estimation, other than the fact that the chosen incidence-based Rt estimates are truncated 7 days earlier to avoid dealing with right censoring, whereas the Ct-based method includes data up to the current day. However, incidence-based Rt methods can deal with right-censoring by including a nowcasting/forecasting component (eg. Abbott & Hellewell et al. Wellcome Open Research 2021 <https://doi.org/10.12688/wellcomeopenres.16006.2>), so it is not quite a fair comparison. It also looks like the incidence-based method panels in Fig 2 are from fitting to the entire dataset and then sub setting by the relevant time windows, rather than re-fitting the Rt model using data that *would* have been available on a given date. If the authors stand by the claim that the Ct method is not affected by right censoring but the incidence-based method is, then I encourage the authors to explain the intuition and to consider if the comparison shown here is fair.

Response: Thank you for your comment. Our Ct-based method works via constructing a link between population Ct distribution and daily incidence-based R_t . Particularly, the Ct-based method can estimate R_t on the current day using samples that were collected on the same day, while the incidence-based methods need to wait for cases who were infected on the current day to become PCR-detectable and reported (i.e., latent period plus reporting delay, which is normally 7 days in Hong Kong as we truncated).

We noted that the nowcasting/forecasting component of incidence-based methods in the mentioned studies were mainly based on partial information of detected cases or extrapolation of recent R_t estimations, which are more of indicative values other than providing genuine real-time estimates as acknowledged by their authors⁹. Whereas we provided the estimations of R_t , using the temporal information that contained in the daily distribution of population viral loads and its established association between incidence-based R_t . We believe this is the most advantageous point of our method.

In our study, incidence-based R_t (as estimated from conventional methods) was taken as the reference for assessing the accuracy of Ct-based R_t , so we used the incidence-based R_t that were fitted to all cases over our study periods. We retrospectively compared our archived real-time estimates along with epidemic progressing (<https://covid19.sph.hku.hk>) with those estimated using the entire data set (as used in our study), and found sub-setting did not lead to divergent results during our study period.

To address your comment and make the advantage of our method clearer, we updated our Fig. 2a-c by truncating the incidence-based R_t , and relevant discussion was added in lines 173-181:

“Conventionally, the main challenge in estimating R_t in real-time was largely caused by the delays between an individual being infected and being PCR detectable or illness onset^{6,19}. Linking the incidence-based R_t and the population-level Ct distribution among samples collected on a given day was able to mitigate the right-censoring issue (i.e., missing cases that were infected but not-yet-observed due to the latent period⁶) encountered by incidence-based methods for assessing transmission^{1,6}. Although studies^{3,4} demonstrated nowcasting and projection of incidence-based R_t during the right-censoring time window, these estimates were indicative values rather than genuine estimates informed with real-time empirical data.”

Reviewer2Comm4.2: It is also not clear how these findings support the claim on L163 that “Our simplified Ct-based method also provides an approach for real-time estimation of R_t without requiring intensive surveillance of COVID-19”. Neither sample

size requirements nor simulation settings with limited data are explored here. To support this claim, I suggest that the authors discuss sample sizes and sample collection frequency much more explicitly -- how many samples are required for robust incidence-based R_t estimation versus Ct-based estimation? Support for these claims with studies would be helpful (eg. simulate incidence curves and Ct values and compare simulation estimates using incidence-based and Ct-based R_t estimation with different sample sizes).

Response: Thank you for your insightful comments and your suggestions. As suggested, we performed simulations to compare the performance of our Ct-based method under settings with various testing capacities. Particularly, we explored scenarios with varying detection rates (scenario 3) and with under detection (scenario 4). Our results showed that our Ct-based estimates were consistent with the simulation truth under these scenarios (Supplementary Fig. 10c-d and Supplementary Table 6).

We hope to clarify that limited surveillance efforts did not necessarily mean low case counts; for example, high number of daily cases were recorded in European countries during the epidemic peak even under constrained surveillance capacity¹⁰. In such sense, a higher under detection proportion around the peak would be expected even though an increasing trend of daily case counts is observed, which may be more likely to affect the incidence-based R_t than the Ct-based R_t (i.e., scenario 4).

To address your comment, we added the simulation results in lines 163-168:

"We found that Ct-based R_t can recover the simulation truth when there is limited and changing case detection (Supplementary Fig. 10). Specifically, Ct-based R_t is correlated with the simulation truth under scenarios with varying case detection (i.e. scenario 3; Spearman $\rho = 0.65$, 95% CI: 0.55- 0.73) and with certain degree of under detection (i.e. scenario 4; Spearman $\rho = 0.50$, 95% CI: 0.36-0.62) (Supplementary Fig. 10 c-d; Supplementary Table 6)."

We acknowledged that both incidence-based and Ct-based methods would encounter greater uncertainty with sparse data, usually during the start and end of an epidemic^{1,2}. We compared the prediction performance under different sample sizes (Supplementary Table 4) and found that 30 cases per day would give very consistent predictions. This

was also in line with the higher consistency between Ct-based R_t and the simulation truth after restricting the comparison to days with over 30 samples under our simulation, as shown in Supplementary Table 6.

To address your comment, we included the additional analysis in Supplementary Table 4 and mentioned in lines 148-152:

“In addition, we found that our model predictions were insensitive to the selection of the training period as long as the training period had sufficient samples (e.g., > 30 samples per day as suggested in Supplementary and Table 4) and could reflect changes in both epidemic growth and population Ct distributions.”

In our simplified Ct-based methods, we used viral loads obtained from RT-qPCR tests that were used for case confirmation in routine surveillance to estimate R_t , and therefore the sample collection frequency is on a daily basis, which is also the same for incidence-based methods.

Minor comments

Reviewer2Comm5: - Title: I might replace “transmissibility” with “transmission rates” or just “transmission”. To me, transmissibility is a fundamental property of the virus, not of the population process.

Response: Thank you for your suggestion, and we have replaced “transmissibility” with “transmission rates” or “transmission” whenever suitable in our revised manuscript.

Reviewer2Comm6: - Abstract: the term RT-qPCR does not appear in the abstract, which is crucial to understand what you are referring to with the term “cycle threshold values”. Eg. L35 “measured by RT-qPCR cycle threshold values”.

Response: Thank you for pointing this out, and we have rephrased this in line 36 as well as other places to emphasize on this.

Reviewer2Comm7: - Introduction: I suggest adding a few additional references on R_t estimation, as the current reference list does not reflect the state-of-the-art. For example: Abbott & Hellewell et al. Wellcome Open Research 2021 <https://doi.org/10.12688/wellcomeopenres.16006.2>; Flaxman et al. Nature

2020 <https://doi.org/10.1038/s41586-020-2405-7>. Also a very recent (since the authors submitted this piece) paper by Parag PLOS Comp Biol

2021 <https://doi.org/10.1371/journal.pcbi.1009347>.

Response: Thank you for your suggestion, and we have revised and updated the citation here accordingly (in lines 48-49).

Reviewer2Comm8: - L57: suggest an additional reference Kissler et al. PLOS Biology <https://doi.org/10.1371/journal.pbio.3001333>

Response: Thank you for the suggestion and we have added it in our reference list in line 59.

Reviewer2Comm9: - Introduction: the aims of the paper feel quite weak at the moment, stating only that the aim is to “determine whether inclusion of data on Ct values could allow real-time estimation of transmissibility”. This has already been shown to be possible, and so I urge the authors to elaborate on what this paper adds, eg. using Ct values from symptom-based surveillance rather than random cross-sectional samples as shown previously; allowing Rt to be estimated with less delay than using case counts alone.

Response: Thank you for your comment. We have mentioned in our previous version that we provided empirical data for validating the application of Ct-based methods under non-random surveillance, as retained in lines 187-191 in our revised version. *“In addition, we showed that the daily Ct distribution could be applied for tracking epidemics under a symptom and contact-tracing based setting, such as Hong Kong, providing empirical data to support the hypotheses generated from previous Ct-based studies¹³.”*

We have further clarified this added value of our study (i.e., its realistic application under symptom-based surveillance) in lines 75-78:

“Here, we incorporated Ct values from COVID-19 cases in Hong Kong, a location with intense surveillance and case-finding efforts, to demonstrate that including data on population viral load distribution from a symptom-based surveillance could achieve real-time tracking of transmission.”

The real-time potential of our Ct-based methods have been demonstrated more thoroughly in our revised manuscript (especially in Fig. 2), as we replied to your previous response to *Reviewer2Comm4.1*.

Reviewer2Comm10: - A key condition in Hay & Kennedy-Shaffer et al. was the use of random cross-sectional samples. It is less clear how samples are obtained here in the main text, though I note that this is explained nicely in the methods. I would suggest i) moving the details on the sample makeup (ie. proportion of symptomatics etc) to the Results section, and ii) adding to the discussion how sample collection strategy impacts how Ct values can be used.

Response: Thank you for your suggestions, and we have merged this information on sampling collection strategies in Results section in lines 94-97:

“A total of 8646 local COVID-19 cases were detected during periods studied, among which 77% (n = 6700) were symptomatic. Asymptomatic cases were more likely to be epidemiologically linked with other known cases compared to symptomatic cases (81% vs. 61%, chi-squared test $p < 0.001$).”

The impact of sampling strategies as indicated by the varying proportion of symptomatic cases over time was explored in Supplementary Fig. 6, which was shown to be insignificant. We have also investigated the impact of varying sampling strategies using simulation, as shown in lines 165-168:

“Specifically, Ct-based R_t is correlated with the simulation truth under scenarios with varying case detection (i.e. scenario 3; Spearman $\rho = 0.65$, 95% CI: 0.55- 0.73) and with certain degree of under detection (i.e. scenario 4; Spearman $\rho = 0.50$, 95% CI: 0.36-0.62) (Supplementary Fig. 10 c-d; Supplementary Table 6).”

We also expanded the discussion (in lines 202-207) to address this comment:

“For example, our results showed that population-level Ct distributions remained informative in tracking epidemic changes over time regardless of changes in surveillance in Hong Kong, especially by expanding the testing capacity during the fourth wave in our case¹⁵. This was further supported by the high accuracy of Ct-based R_t under various settings of case detection in our simulations.”

Reviewer2Comm11: - What was the gene target used to generate the Ct values?

Response: Thank you for your comment, and E gene was used in RT-qPCR tests to generate the Ct values in Hong Kong. We mentioned this information in both main text (line 87-90) and the Methods section (lines 299-300).

“After excluding imported cases, we analyzed the first available record of Ct value (derived from RT-qPCR tests targeting E gene¹⁶) for each confirmed case and characterized the daily distribution of Ct values (measured by mean and skewness) that were sorted by sampling days.”

“Results for SARS-CoV-2 RT-qPCR tests (LightMix® Modular SARS-CoV-2 (COVID-19) E-gene, TIB Molbiol/Roche, Berlin, Germany¹⁶) were recorded as Ct values in the system.”

Reviewer2Comm12: - Fig 2: make clearer in panels a-d which data are available in each window (eg. add a vertical dashed line or truncate the case counts).

Response: Thank you for your suggestion, and we have updated the Fig. 2 by truncating case counts and pointing out the date of estimation in panel a; we also truncated the incidence-based R_t in panel b-c to illustrate this point.

Reviewer2Comm13: - L147: what is the basis for calling these estimates precise? Many of the confidence intervals seem far wider than the incidence-based R_t estimates.

Response: Thank you for your comment. We compared the correlation between the estimated mean of R_t from both the incidence-based and Ct-based methods using our empirical example in Fig. 2d (also in lines 132-135). We found that the Spearman correlation ρ between R_t estimated from the two methods was 0.81 ($p < 0.001$) and 0.48 ($p < 0.001$) respectively for training and testing periods, suggesting good accuracy of predictions.

“We found high correlations between Ct- and incidence-based R_t for both training (Spearman’s correlation coefficient, $\rho = 0.81$, $P < 0.001$) and testing periods ($\rho = 0.48$, $P < 0.001$) (Fig. 2b-d).”

We think the wide interval of Ct-based R_t is more likely to be affected to the analytic methods rather than the inaccuracy of the estimation. We provided prediction intervals for Ct-based R_t , which by definition is wider than confidence interval. In addition, our

Ct-based R_t estimation implicitly assumed a 1-day smoothing window, which would give a larger uncertainty compared to a 14-day smoothing window (which is usually used in incidence-based methods¹).

Reviewer2Comm14: - L154: I would argue that the cited methods provide longitudinal growth rate estimates, which are comparable to R_t given that growth rate and reproductive number can be linked using knowledge of the natural history of SARS-CoV-2.

Response: Thank you for this comment and we have rephrased the sentence to avoid misconception (in line 178):

“Linking the incidence-based R_t and the population-level Ct distribution among samples collected on a given day was able to mitigate the right-censoring issue (i.e., missing cases that were infected but not-yet-observed due to the latent period⁶) encountered by incidence-based methods for assessing transmission^{1,6}.”

Reviewer2Comm15: - Fig 2 and S1: it would be useful to have some indication of sample size for in each of these time windows. For example, are the very differing R_t estimates in 2c driven by very small Ct sample sizes?

Response: Thank you for your comment. The difference between R_t estimates from the two methods in Fig. 2c were indeed driven by the very few Ct samples when the circulation was low. We have added daily case counts for each panel in Fig. 2 to demonstrate the daily sample size.

To further address your comment, we also examined the impact of sample sizes on the consistencies of R_t estimations in Supplementary Table 4 and in lines 413-417:

“For sample sizes, we calculated the Spearman correlation coefficients between incidence-based and Ct-based estimates under different sample size intervals, and found that Ct-based R_t tended to be more accurate with over 30 records per day (Supplementary Table 4).”

Reviewer2Comm16: - L187 typo “chances”.

Response: Revised, thank you.

Reviewer2Comm17: - L381: typo “rest nine sets”

Response: Corrected, thank you.

References:

- 1 Cori, A., Ferguson, N. M., Fraser, C. & Cauchemez, S. A New Framework and Software to Estimate Time-Varying Reproduction Numbers During Epidemics. *American Journal of Epidemiology* **178**, 1505-1512, doi:10.1093/aje/kwt133 (2013).
- 2 Gostic, K. M. *et al.* Practical considerations for measuring the effective reproductive number, Rt. *PLOS Computational Biology* **16**, e1008409, doi:10.1371/journal.pcbi.1008409 (2020).
- 3 Kucirka, L. M., Lauer, S. A., Laeyendecker, O., Boon, D. & Lessler, J. Variation in False-Negative Rate of Reverse Transcriptase Polymerase Chain Reaction-Based SARS-CoV-2 Tests by Time Since Exposure. *Annals of Internal Medicine* **173**, 262-267, doi:10.7326/M20-1495 (2020).
- 4 Jones, T. C. *et al.* Estimating infectiousness throughout SARS-CoV-2 infection course. *Science* **373**, eabi5273, doi:10.1126/science.abi5273 (2021).
- 5 Néant, N. *et al.* Modeling SARS-CoV-2 viral kinetics and association with mortality in hospitalized patients from the French COVID cohort. *Proc Natl Acad Sci U S A* **118**, doi:10.1073/pnas.2017962118 (2021).
- 6 Hay, J. A. *et al.* Estimating epidemiologic dynamics from cross-sectional viral load distributions. *Science*, eabh0635, doi:10.1126/science.abh0635 (2021).
- 7 Chen, P. Z. *et al.* Heterogeneity in transmissibility and shedding SARS-CoV-2 via droplets and aerosols. *eLife* **10**, e65774, doi:10.7554/eLife.65774 (2021).
- 8 Walker, A. S. *et al.* Ct threshold values, a proxy for viral load in community SARS-CoV-2 cases, demonstrate wide variation across populations and over time. *eLife* **10**, e64683, doi:10.7554/eLife.64683 (2021).
- 9 Abbott, S. *et al.* Estimating the time-varying reproduction number of SARS-CoV-2 using national and subnational case counts [version 2; peer review: 1 approved with reservations]. *Wellcome Open Research* **5**, doi:10.12688/wellcomeopenres.16006.2 (2020).
- 10 Pullano, G. *et al.* Underdetection of cases of COVID-19 in France threatens epidemic control. *Nature* **590**, 134-139, doi:10.1038/s41586-020-03095-6 (2021).

REVIEWER COMMENTS

Reviewer #1 (Remarks to the Author):

I appreciate the authors' efforts in revising the manuscript. The additional simulation studies allowed detailed inspection of the method and provided stronger support for the results. Questions regarding real-world applications were also addressed in the revision.

Reviewer #2 (Remarks to the Author):

I am satisfied that the authors have addressed most of my (reviewer 2) comments, and I think the new simulation-recovery scenarios really strengthen the paper. The findings are a valuable contribution to the literature and now have robust sensitivity analyses and discussion to support the claims. I have three remaining comments which should be addressed, but otherwise the paper is in good shape to be published.

1. Importance of the onset-to-sampling delay distribution

First, regarding Reviewer2Comm3.2, where I think my initial comment could have been much clearer. I am not fully convinced that these results show epidemic progress alone as a better explanation of changes in population Cts than changes in testing delays. I think we are almost on the same page from the response to Reviewer2Comm3.2, but my point is that the *observed* distribution of onset-to-sampling times changes over time due to epidemic dynamics, irrespective of changes in testing protocols/patterns.

Consider Figure S1 – it looks like the Ct value and the onset-to-sampling delay distributions mirror each other. This is consistent with the fact that the distribution of observed sampling delays is the convolution of incidence and the forward sampling delay distribution – as shown in reference 1 (Gostic et al.) and 13 (Hay et al. Fig S3&4). *Observed* sampling delays therefore change over time without requiring any change in detection patterns. Thus, this suggests that we might be able to just use the onset-to-sampling delay distribution to train the model and infer Rt, rather than needing the Ct values. Or put another way – with a dataset such as this, does including the Ct distributions add information beyond just using the onset-to-sampling delay distributions? What if Ct mean/skew were replaced with onset-to-sampling mean/skew in Eq. 7? However, I do note that including this variable might not affect the explanatory power of the overall model and acknowledge that this isn't a causal inference study.

For example, consider two trivial directed acyclic graphs:

1. Rt -> Ct value distribution
2. Rt -> onset delay distribution -> Ct value distribution

In (2), if we know the onset delay distribution, then we do not gain anything from knowing the Ct value distribution ie. we have a pipe.

This does not diminish the value of using Ct values for epi inference – there are many scenarios where the time-since-onset delay is not reported or known, and the full DAG may be more complicated than the one shown above (so knowing Cts does add information on top of knowing sampling delays). But I think this is important to interpret the mechanism underpinning the relationship shown here.

If the authors agree that this argument applies (and it might not), I would:

1. See if modifying Eq 7 to use onset-to-sampling delays instead of Ct values still provides a useful predictive model relative to the Ct model (eg. compared by variation explained).
2. Update the discussion to make clear to the reader that the causal structure is not as simple as "Rt -> Cts", but that including Cts may be useful for inference nonetheless.

Simulation results

I think the simulation-recovery experiments are a fantastic addition. However, Figure S10 does

raise some concerns, as the Ct-based method appears to be systematically biased upward at the tail end of the second wave, particularly for scenarios 3 and 4. The estimates in scenario 4 are pretty wild in general. Please discuss potential reasons for and implications of this systematic bias.

Further clarification on reason for sample collection

Thank you for including additional information on the symptomatic/asymptomatic proportion of samples in response Reviewer2Comment10. I would suggest adding information, or at least discussing, that a case being "asymptomatic" could mean a few things. It may be that the asymptomatic individual is tested at random (thus unknown time-since-infection), or because of suspected recent exposure (thus skewed towards early time-since-infection), which will impact their expected Ct value.

Reviewer #1 (Remarks to the Author):

Reviewer1Comm1: I appreciate the authors' efforts in revising the manuscript. The additional simulation studies allowed detailed inspection of the method and provided stronger support for the results. Questions regarding real-world applications were also addressed in the revision.

Response: Thank you for your time and supportive comment.

Reviewer #2 (Remarks to the Author):

Reviewer2Comm1: I am satisfied that the authors have addressed most of my (reviewer 2) comments, and I think the new simulation-recovery scenarios really strengthen the paper. The findings are a valuable contribution to the literature and now have robust sensitivity analyses and discussion to support the claims. I have three remaining comments which should be addressed, but otherwise the paper is in good shape to be published.

Response: Thank you for your supportive comment, and we have provided our point-by-point responses to your remaining comments below.

Reviewer2Comm2: 1. Importance of the onset-to-sampling delay distribution
First, regarding Reviewer2Comm3.2, where I think my initial comment could have been much clearer. I am not fully convinced that these results show epidemic progress alone as a better explanation of changes in population Cts than changes in testing delays. I think we are almost on the same page from the response to Reviewer2Comm3.2, but my point is that the *observed* distribution of onset-to-sampling times changes over time due to epidemic dynamics, irrespective of changes in testing protocols/patterns.

Consider Figure S1 – it looks like the Ct value and the onset-to-sampling delay distributions mirror each other. This is consistent with the fact that the distribution of observed sampling delays is the convolution of incidence and the forward sampling delay distribution – as shown in reference 1 (Gostic et al.) and 13 (Hay et al. Fig S3&4). *Observed* sampling delays therefore change over time without requiring any change in detection patterns. Thus, this suggests that we might be able to just use the onset-to-

sampling delay distribution to train the model and infer R_t , rather than needing the C_t values. Or put another way – with a dataset such as this, does including the C_t distributions add information beyond just using the onset-to-sampling delay distributions? What if C_t mean/skew were replaced with onset-to-sampling mean/skew in Eq. 7? However, I do note that including this variable might not affect the explanatory power of the overall model and acknowledge that this isn't a causal inference study.

For example, consider two trivial directed acyclic graphs:

1. $R_t \rightarrow C_t$ value distribution
2. $R_t \rightarrow$ onset delay distribution $\rightarrow C_t$ value distribution

In (2), if we know the onset delay distribution, then we do not gain anything from knowing the C_t value distribution ie. we have a pipe.

This does not diminish the value of using C_t values for epi inference – there are many scenarios where the time-since-onset delay is not reported or known, and the full DAG may be more complicated than the one shown above (so knowing C_t s does add information on top of knowing sampling delays). But I think this is important to interpret the mechanism underpinning the relationship shown here.

If the authors agree that this argument applies (and it might not), I would:

1. See if modifying Eq 7 to use onset-to-sampling delays instead of C_t values still provides a useful predictive model relative to the C_t model (eg. compared by variation explained).
2. Update the discussion to make clear to the reader that the causal structure is not as simple as " $R_t \rightarrow C_t$ ", but that including C_t s may be useful for inference nonetheless.

Response: Thank you for your insightful comment and sorry for the confusion here. We agree with you that the onset-to-sampling delay would change with the epidemic trajectory - with or without the impact of testing patterns – under symptom-based surveillance. Therefore, we also agree that it is important to include the observed onset-to-sampling distribution of into our model. We have previously mentioned the model

performance by including the delay terms of onset-to-sampling in the manuscript in lines 440-444:

“We also adjusted for delays from illness onset to sampling in our main model (Eq. 7) and found that changes in sample collections (coefficient $\beta = 0.93$, 95% CI: 0.87-1.01) did not alter the association between population Ct distribution and incidence-based R_t (Supplementary Table 5).”

To test if the onset-to-sampling distribution alone would predict R_t in our empirical example, we further compared prediction models that used population viral loads and/or onset-to-sampling delays in more detail as below (also included as Supplementary Table 5). We found that models that only included the observed onset-to-sampling delay explained much lower variance compared to our main model, suggesting that the onset-to-sampling delay alone was not be able to predict R_t . In addition, further inclusion of the onset-to-sampling delay to our main model did not explain more variance in epidemic changes compared to the main model in our example.

Training period	Main (6 Jul to 31 Aug 2020)		Alternative (20 Nov to 19 Dec)	
	β (95% CI)	Adjusted R square	β (95% CI)	Adjusted R square
Ct alone (main model)		0.72		0.71
Ct mean	0.86 (0.81, 0.92)		0.89 (0.83, 0.95)	
Ct skewness	1.75 (1.11, 2.75)		1.57 (1.10, 2.24)	
Ct and delay		0.72		0.73
Ct mean	0.87 (0.82, 0.93)		0.92 (0.85, 0.99)	
Ct skewness	1.65 (0.99, 2.75)		1.52 (1.07, 2.17)	
Delay mean	0.93 (0.87, 1.01)		0.93 (0.80, 1.08)	
Delay skewness	1.00 (0.84, 1.18)		1.11 (0.97, 1.27)	
Onset-to-sampling delay alone		0.26		0.54
Delay mean	0.79 (0.71, 0.88)		0.72 (0.64, 0.81)	
Delay skewness	1.08 (0.84, 1.38)		1.16 (0.98, 1.37)	

The fundamental assumption of Ct-based methods is that the epidemic trajectory can be inferred from population viral load distribution that changes with infection-to-sampling delay, which is the (often not observable) infection-to-onset (i.e., incubation period) and (observable) onset-to-sampling. In a random sampling scenario, it is mostly the

distribution of infection-to-onset to inform the collective exposure time, as the onset-to-sampling delay could be randomized. However, under symptom-based surveillance, onset-to-sampling distribution may or may not affect Ct values and therefore its impact on observed Ct distributions need to be examined. In our case, Ct distributions (the proxy for collective exposure time) still existed even after controlling for the onset-to-sampling delay (Table above), suggesting that our findings on the association between population Ct value distribution and the transmission dynamics was mainly driven by the changing distribution of delays between infection and symptom onset (Fig. S4 in Hay J *et al.*, 2021 *Science*¹).

To address the comment, we now included the analysis that added onset-to-sampling delay to our main model as suggested (Supplementary Table 5). We also discussed the possible underlying mechanism in lines 194-199:

“Temporal changes in population Ct distribution over an epidemic largely reflect changes in infection-to-sampling delays that is determined by delays of infection-to-onset (i.e., incubation) and onset-to-testing (i.e., testing delay). We showed that the onset-to-testing did not provide additional information to the population Ct distribution (Supplementary Table 5), suggesting the observed changes in Ct distribution was largely driven by collective changes in the exposure time.”

Reviewer2Comm3: Simulation results

I think the simulation-recovery experiments are a fantastic addition. However, Figure S10 does raise some concerns, as the Ct-based method appears to be systematically biased upward at the tail end of the second wave, particularly for scenarios 3 and 4. The estimates in scenario 4 are pretty wild in general. Please discuss potential reasons for and implications of this systematic bias.

Response: Thank you for your comment. We carefully checked our choices of parameters for the Ct trajectory model and noticed that the duration of viral shedding under our chosen parameter should be counted from symptom onset², which has, however, been parameterized as the time since exposure in a previous study³ as well as in our previous analysis. This has led to unreasonable trajectories for some cases in the simulation with long incubation periods (i.e., rebound of viral load if the incubation period is longer than the simulated durations of shedding), which subsequently led to

overestimation of the simulated viral load at sampling for these cases. Such overestimation had an exaggerated effects on R_t estimation during the tail of the epidemic, as we are more likely to detect cases with longer incubation periods at the tail of the epidemic (which is the underlying assumption of the Ct-based methods).

We have therefore updated this specific parameter to count the duration of viral shedding from onset (as revised in lines 510-512) and revised relevant simulation results (Supplementary Figs. 9-10 and Supplementary Table 7), which have solved this issue and also improved accuracies of our estimates. We have also revised in lines 168-171 in our manuscript accordingly.

“The duration of viral shedding since onset was parameterized as normally distributed with mean and SD of 17 and 0.94 days³⁸.”

“Specifically, Ct-based R_t is correlated with the simulation truth under scenarios with varying case detection (i.e., scenario 3; Spearman $\rho = 0.77$, 95% CI: 0.73, 0.81) and with certain degree of under detection (i.e., scenario 4; Spearman $\rho = 0.65$, 95% CI: 0.53, 0.75) (Supplementary Fig. 10 c-d; Supplementary Table 7).”

As we employed 1-day smoothing window to derive daily Ct distributions for R_t estimation, the estimates will be less smooth as compared with the incidence-based R_t which have used a 14-day smoothing window. We could see from Figure 3 (see below) in Cori et.al⁴ that incidence-based R_t could also vary a lot at the initial start of an epidemic under 1-day smoothing window due to larger uncertainty caused by low incidences at that time point.

[REDACTED]

From their Figure 3 panel C⁴ where the performance of the 1-day smoothing window was demonstrated, we could barely see any obvious trend throughout, whereas our Ct-

based R_t could become stable shortly after acting wildly at the start and did not perform worse in such sense.

We therefore believe the selection of smoothing window is the main reason for the varied estimates at the start of the simulated second wave in scenario 4 where larger uncertainty existed due to fewer daily samples available due to under detection.

Reviewer2Comm4: Further clarification on reason for sample collection

Thank you for including additional information on the symptomatic/asymptomatic proportion of samples in response Reviewer2Comment10. I would suggest adding information, or at least discussing, that a case being “asymptomatic” could mean a few things. It may be that the asymptomatic individual is tested at random (thus unknown time-since-infection), or because of suspected recent exposure (thus skewed towards early time-since-infection), which will impact their expected Ct value.

Response: Thank you for your suggestion. In Hong Kong, asymptomatic cases is defined as cases without symptoms at the time of testing, and whether these cases subsequently developed symptoms were not reported. According to this definition, asymptomatic cases in Hong Kong could also include pre-symptomatic detections of infection. We further looked at the detection sources of asymptomatic cases included in our study, and found that 81% (1584 out of 1946) were contacts of known local cases, suggesting the contact tracing is likely the source of detection of these cases. The remaining 19% asymptomatic cases that were unlinked to any other cases were mostly detected via community testing that targeted at populations with predefined high risks of exposures (which were more like testing at random).

We compared the distribution of Ct values for both linked and unlinked asymptomatic cases with the distribution for symptomatic cases (see the figure below). Despite for an eyeballing trend, we found no significant difference in Ct values distribution between asymptomatic cases (both linked and unlinked) and symptomatic cases.

We agree with the reviewer that a discussion about indications of asymptomatic cases would facilitate interpretation of our methods, while we have demonstrated trivial impact of asymptomatic cases on R_t estimations in our case (Supplementary Fig. 6). This is likely due to the high proportion of symptomatic cases within a symptom-based surveillance setting, which could lead to even higher proportions of symptomatic cases detected in places outside Hong Kong where intense contact tracing or large-scale testing has not been implemented.

To address the comment, we extended relevant results (lines 95-100) and discussion (lines 211-215) about the asymptomatic cases:

“Cases who were asymptomatic at the time of testing were more likely to be epidemiologically linked with other known cases compared to symptomatic cases (81% vs. 61%, chi-squared test $p < 0.001$), suggesting they were more likely to be detected from contact tracing or from compulsory testing for populations with predefined high risks of exposures (see Methods) and could be detected earlier than symptomatic cases.”

“For example, our results showed that population-level Ct distributions remained informative in tracking epidemic changes over time regardless of changes in surveillance in Hong Kong, especially by expanding the testing capacity and therefore detecting more cases at earlier disease stage (i.e., asymptomatic cases; Supplementary Fig. 6) during the fourth wave in our case¹⁵.”

References:

- 1 Hay, J. A. *et al.* Estimating epidemiologic dynamics from cross-sectional viral load distributions. *Science*, eabh0635, doi:10.1126/science.abh0635 (2021).
- 2 Cevik, M. *et al.* SARS-CoV-2, SARS-CoV, and MERS-CoV viral load dynamics, duration of viral shedding, and infectiousness: a systematic review and meta-analysis. *The Lancet Microbe* **2**, e13-e22, doi:10.1016/S2666-5247(20)30172-5 (2021).
- 3 Quilty, B. J. *et al.* Quarantine and testing strategies in contact tracing for SARS-CoV-2: a modelling study. *The Lancet Public Health* **6**, e175-e183, doi:10.1016/S2468-2667(20)30308-X (2021).
- 4 Cori, A., Ferguson, N. M., Fraser, C. & Cauchemez, S. A New Framework and Software to Estimate Time-Varying Reproduction Numbers During Epidemics. *American Journal of Epidemiology* **178**, 1505-1512, doi:10.1093/aje/kwt133 (2013).